# Effects of short-term arm immobilization on motor skill acquisition

**Erin M. King** [1,2], **Lauren L. Edwards**[2], **Michael R. Borich**[2,3]*

**1** Neuroscience Graduate Program, Graduate Division of Biological and Biomedical Sciences, Emory University, Atlanta, GA, United States of America, **2** Division of Physical Therapy, Department of Rehabilitation Medicine, Emory University, Atlanta, GA, United States of America, **3** Department of Biomedical Engineering, Georgia Institute of Technology, Atlanta, GA, United States of America

* michael.borich@emory.edu

**Data Availability Statement:** Raw behavioral data are available from the Open Science Framework database (link to view: https://osf.io/wazdy/?view_only=cfed7648613f45589c86d73582f93ce7).

**Funding:** EMK is supported by the Emory Mechanisms of Learning Across Development and

## Abstract

Learning to sequence movements is necessary for skillful interaction with the environment. Neuroplasticity, particularly long-term potentiation (LTP), within sensorimotor networks underlies the acquisition of motor skill. Short-term immobilization of the arm, even less than 12 hours, can reduce corticospinal excitability and increase the capacity for LTP-like plasticity within the contralateral primary motor cortex. However, it is still unclear whether short-term immobilization influences motor skill acquisition. The current study aimed to evaluate the effect of short-term arm immobilization on implicit, sequence-specific motor skill acquisition using a modified Serial Reaction Time Task (SRTT). Twenty young, neurotypical adults underwent a single SRTT training session after six hours of immobilization of the non-dominant arm or an equivalent period of no immobilization. Our results demonstrated that participants improved SRTT performance overall after training, but there was no evidence of an effect of immobilization prior to task training on performance improvement. Further, improvements on the SRTT were not sequence-specific. Taken together, motor skill acquisition for sequential, individuated finger movements improved following training but the effect of six hours of immobilization was difficult to discern.

## 1. Introduction

Learning to coordinate motor sequences is essential for completing tasks necessary for daily living, such as tying a shoe or typing. Experience is a potent driver of neuroplasticity throughout the cortex [1], and even short-term experiences, such as motor skill practice, have been shown to induce structural and functional changes within the human motor cortex and to underlie motor learning [2, 3]. Synaptic processes such as long-term potentiation (LTP) and long-term depression (LTD) [4], alterations of dendritic spine density and morphology [5], and changes in inhibitory neurotransmission [3, 6] have all been shown to contribute to the training-induced plasticity that underlies motor skill acquisition.

Previous research in humans has demonstrated that experience-dependent synaptic strengthening through LTP-like plasticity in sensorimotor circuits is necessary for acquisition

Species Training Grant 2T32HD071845-06 (https://sites.google.com/view/mechanismsoflearning/home), a Ruth L. Kirschstein Institutional National Research Service Award through the National Institutes of Health. The funders had no role in study design, data collection and analysis, decision to publish, or preparation of the manuscript.

**Competing interests:** The authors have declared that no competing interests exist.

of motor skill [3]. However, in order to maintain stable levels of activity in these circuits, it is necessary to regulate synaptic strength at the level of the individual synapse to maintain synaptic homeostasis [7]. The model of homeostatic plasticity suggests that the degree of experience-dependent strengthening or weakening that can occur in a given synapse is influenced by the recent history of synaptic activity [8, 9], which prevents the circuit from becoming over- or under-excited [1, 10–12]. For example, in a synapse that has recently undergone a period of synaptic strengthening, additional synaptic strengthening becomes less likely to occur, and synaptic weakening becomes more likely. This principle has been demonstrated in both excitatory and inhibitory circuits within M1 using noninvasive brain stimulation (NIBS), where increasing activity of these circuits prior to plasticity-induction protocols resulted in reduced potential for further LTP-like plasticity [13, 14]. Since LTP is a primary contributor to experience-dependent plasticity, inducing LTD-like plasticity within M1 prior to training may leverage homeostatic mechanisms to enhance the capacity for task-specific synaptic strengthening and performance improvement in humans.

Short-term ($\leq$12hr) limb immobilization transiently reduces sensory input to and motor output from the contralateral sensorimotor cortex, resulting in a temporary decrease in corticospinal excitability [15, 16], thought to reflect a decrease in synaptic strength through LTD-like processes [17], as well as EEG markers of synaptic potentiation [17]. Further, the capacity for LTP-like plasticity is enhanced immediately after short-term immobilization in human primary motor cortex [15]. Although short-term limb immobilization modifies systems-level indices of synaptic strength in humans, the effect on motor skill training-induced plasticity is unclear. Given that plasticity within human M1 underlies sensorimotor skill learning [3, 9, 18], an intervention that has the potential to enhance the capacity for LTP-like plasticity in M1 may influence skill learning. While several studies have found that motor performance on a variety of tasks is decreased after short-term immobilization [16, 17, 19–24], only one has examined the effect of immobilization on skill acquisition [22]. A study by Opie and Evans found no clear effect of immobilization on training during a grooved pegboard task [22]. However, no studies have examined the effect of immobilization on a task that requires individuated, sequenced finger movements. The purpose of the current study was to evaluate the effects of short-term limb immobilization on implicit motor skill acquisition in young, healthy individuals. In this study, participants completed a single motor skill training session after a period of 6 hours with or without immobilization of the non-dominant arm. We hypothesized that if short-term limb immobilization increases the capacity for activity-dependent synaptic strengthening in the corresponding contralateral M1 representation, then greater motor skill acquisition would be observed with training that followed immobilization compared to training following an equivalent period of no immobilization.

## 2. Materials and methods

### 2.1 Study participants

21 healthy individuals (6 male) aged 18–35 (24.8 ± 4.7 years) participated in the current study spanning the morning and evening of one day. The age range was selected in order to reduce the influence of age on motor skill learning [25]. Inclusion criteria included (1) no history of movement impairment or neurodegenerative disease and (2) right handedness according to the Edinburgh Handedness Scale [26]. All study procedures were approved by the Emory University Institutional Review Board in accordance with the Declaration of Helsinki. Written consent was obtained from all participants prior to testing procedures. Behavioral data from one participant were excluded due to equipment malfunction.

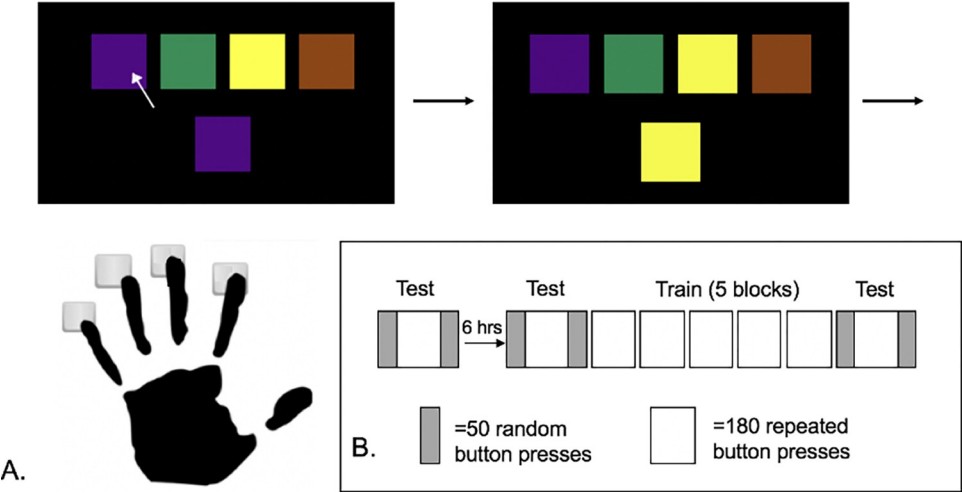

**Fig 1. Modified Serial Reaction Time Task (SRTT).** (A) Participants were instructed to press the key on a custom button box that corresponded to the top square that matched the target (bottom) square (B) Sequence-specific skill (SKILL) was calculated as the mean difference in response time (RT) between repeated (white) and random (gray) sequences for test blocks at each time point.

## 2.2 Motor task paradigm

A version of the Serial Reaction Time Task (SRTT) [27, 28], an implicit motor task, modified from Clos and Sommer was used in this study [29]. Participants placed the fingers of their left (non-dominant) hand on a custom-made button box. The buttons on the box corresponded to the top four colored squares on a computer display placed at eye-level in front of the participant (**Fig 1**). Participants were instructed to press the button corresponding to the top square that matched the color of the bottom (target) square as quickly and accurately as possible. An example trial and overall study design are shown in **Fig 1.** In the morning baseline (BL) session, participants completed one test block that consisted of 280 button presses (50 random, 180 repeated, 50 random) during the session prior to the immobilization period to assess motor performance [30, 31]. Repeated button presses consisted of 15 repeats of a 12-item sequence, and the participants were not informed there was a repeating sequence.

## 2.3 Arm immobilization

After completion of the BL session, 10 participants were randomly assigned to undergo arm immobilization. These individuals were instructed to wear a finger control mitt on the left (non-dominant) hand, which secured positioning of the fingers in a padded mitt to restrict finger movement. In addition to the hand mitt, the arm was placed in a sling to reduce movement of the wrist, elbow, and shoulder joints. Participants were instructed to move their left arm as little as possible during the immobilization period but to use the non-immobilized right (dominant) hand and arm normally. Individuals in the control group (n = 10) were instructed to use both arms normally between sessions.

## 2.4 Motor skill acquisition paradigm

After the 6-hour immobilization period, skill acquisition was assessed in the evening using a pre-train test block (PRE), five training blocks consisting of 180 repeated button presses, and a post-train test block (POST). Colors changed each trial in order to mask the sequence and prevent explicit awareness of the sequence. At the end of the evening session, the degree of explicit

awareness of the presence of a sequence was assessed by asking if the participants noticed any pattern of button presses during the task. If they indicated that they noticed a pattern during training, they were asked to freely recall the sequence. The ability to freely recall ≥4 consecutive items of the 12-item sequence was considered explicit awareness [30, 31].

Raw data files as well as a summary data file can be found here: https://osf.io/wazdy/?view_only=cfed7648613f45589c86d73582f93ce7

## 2.5 Behavioral outcome measures

During the six-hour immobilization period, activity monitors (wGT3x-BT, ActiGraph) were worn on both arms by all participants to determine compliance with the immobilization procedure since participants were allowed to leave the lab between test sessions. Movement of each arm, measured in units of Gs, was collected bilaterally (left/target and right/non-target arms) throughout the six-hour immobilization period for both groups. A two-way ANOVA was performed to examine the effect of immobilization on activity counts, with within-subject factor of hand (two levels: target and non-target) and between-subject factor of group (two levels: immobilization or control group).

**2.5.1 Assessment of general motor performance.** Response time for each button press was acquired during task performance with a custom Java script. Outlier response times, defined as a response time three standard deviations greater than the mean response time within each block for each participant, were removed from analysis.

General motor performance was assessed by calculating the response times for button presses across task exposure. Response times were then normalized to the average response time for the first 50 random button presses in order to account for variations in baseline motor performance. Normalized response times for random sequence in the test blocks and repeated sequence in the training blocks were analyzed separately. A two-way ANOVA was used to assess the effect of immobilization on general motor performance, as measured by the normalized response time for the first 50 random sequence button presses during each of the three test blocks, with within-subject factor of time (three levels: baseline (BL), pre-training (PRE), and post-training (POST)) and between-subject factor of group (two levels: immobilization or control group). A separate two-way ANOVA was used to assess normalized response time for repeated sequence performance across the training blocks (five levels: Training blocks 1–5) with between-subject factor of group (two levels: immobilization or control group).

**2.5.2 Assessment of sequence-specific skill.** To assess sequence-specific skill, two outcome measures were calculated: Skill Score (SS) and Interference Score (IS). In line with previous studies using a similar version of the SRTT and task design [30, 31], the Skill Score (SS) was calculated as the average response time for the last 50 random button presses of each test block minus the average response time for the last four repeated sequences (48 repeated button presses) preceding the random presses [32]. This was kept consistent across test blocks to control for potential order effects. A larger SS indicates greater sequence-specific skill, such that when trained, participants' repeated sequence performance is faster than random sequence performance. Skill Scores were calculated for BL ($SS_{BL}$), PRE ($SS_{PRE}$), and POST ($SS_{POST}$) test blocks.

The Interference Score (IS) was calculated to assess potential interference caused by an abrupt transition from repeated sequence to random sequence (Rep-Rand) button presses, which leads to an increase in response time [28]. Rather than assessing the relative response times of repeated sequence and random sequence button presses averaged over many button presses, the IS examines the impact of disrupting the trained motor sequence with a random sequence by calculating the average response time for the first 12 random button presses

immediately following a transition minus the average response time for the 12 repeated button presses immediately preceding the transition. A larger IS represents a larger increase in response time at the transition from random to repeat and therefore greater interference. Interference Score (Rep-Rand) was calculated for each test block ($IS_{BL}$, $IS_{PRE}$, $IS_{POST}$).

**2.5.3 Task performance of the non-immobilized hand.** Previous research has demonstrated that interhemispheric interactions between motor cortices have been shown to change after a period of immobilization [33]. In order to assess possible effects of immobilization of the left hand on the right hand, two test blocks of the SRTT were performed with the right, nontarget hand: one at the BL timepoint and one at the PRE timepoint. Statistics were performed in order to assess the effects of immobilization on both general motor performance as well as sequence-specific skill, although no training blocks occurred between these two test blocks. A two-way ANOVA was used to examine normalized response time for the last 50 random button presses of each test block. A separate two-way ANOVA was used to assess skill score before and after the immobilization period (two levels: BL and PRE blocks) with between-subject factor of group (two levels: immobilization or control group).

All statistical analyses were performed with Prism GraphPad 8, and a critical α was set at 0.05 corrected for multiple comparisons as appropriate. Normality and homogeneity of variance were statistically confirmed with the Shapiro-Wilk's Test, descriptive statistics, and Levene's Test.

## 3. Results

A two-way ANOVA demonstrated that there were significant effects of time (F = 60.1, p < .0001) and of group (F = 14.3, p < .01), as well as a time X group interaction (F = 40.5, p < .0001) on average activity counts (**Fig 2**). Sidak's multiple comparisons test indicated that activity in the immobilized (target) arm were significantly reduced in immobilized individuals

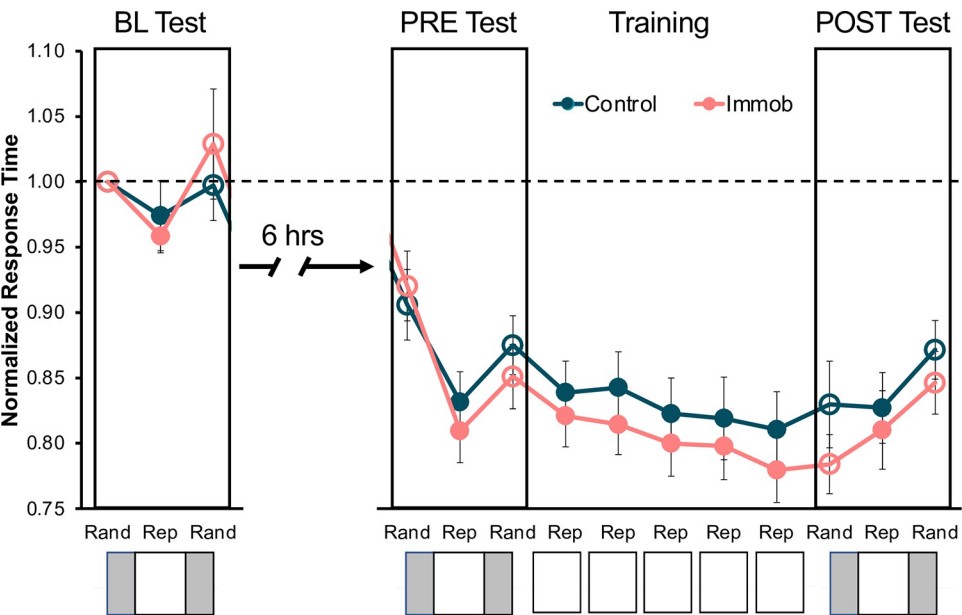

**Fig 2. Immobilized participants complied with the immobilization procedure.** Participants wore activity monitors on both wrists throughout the six-hour immobilization period. Activity counts in the immobilized (target) arm were significantly reduced compared to the non-immobilized (nontarget) limb in immobilized participants (t = 9.98, p < .0001).

**Table 1. Activity counts ANOVA results.**

| Measure | Source | DFn, DFd | F Statistic | p-value | Effect size (eta$^2$) |
|---|---|---|---|---|---|
| Activity Counts | Time | 1, 18 | 60.1 | **< .0001** | .223 |
| | Group | 1, 18 | 14.3 | **< .01** | .248 |
| | Time X Group | 1, 18 | 40.5 | **< .0001** | .151 |

Significant p-values are bolded.

(t = 9.98, p < .0001) confirming participants complied with the immobilization procedure. **Table 1** summarizes the ANOVA results for activity counts.

### 3.1 Assessment of general motor performance

Overall, 1.99% of button presses were removed (694/34,800 total button presses) across all participants (range: 1.38%-2.76%). Average response times were determined to be normally distributed for each block of button presses: BL (W = .95, p = .3), PRE (W = .95, p = .4), TRAIN1 (W = .95, p = .34), TRAIN2 (W = .95, p = .38), TRAIN3 (W = .97, p = .74), TRAIN4 (W = .97, p = .78), TRAIN5 (W = .94, p = .21), and POST (W = .98, p = .95). Our results demonstrate that general motor performance improved across task exposure, measured by a decrease in response time for both random and repeated button presses with task exposure. The overall accuracy for control participants was 97.4%, and the overall accuracy for immobilized participants was 96.4%. Normalized response times for each group across training are shown in **Fig 3**.

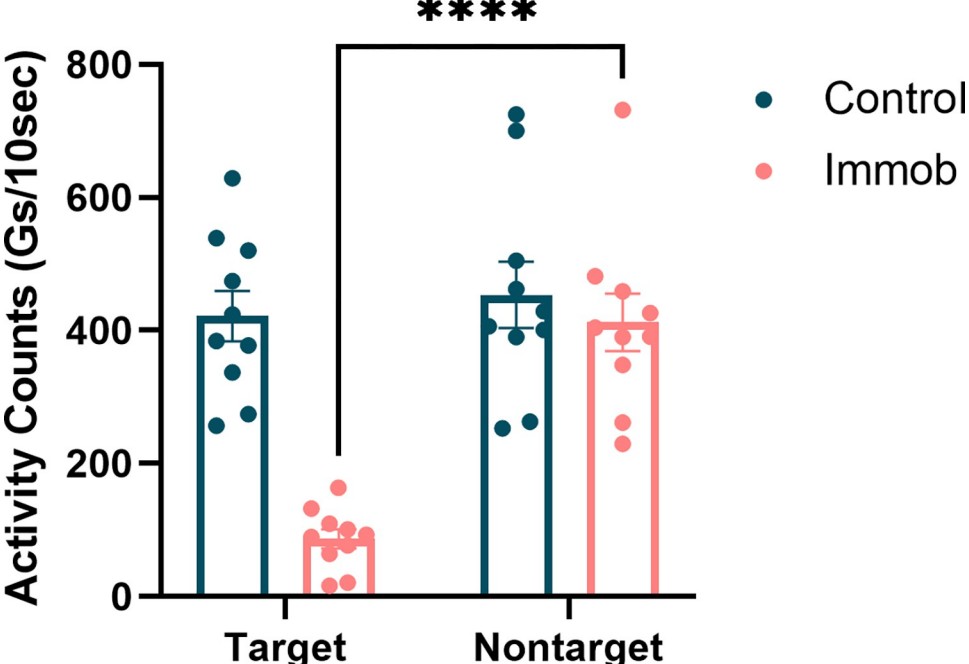

**Fig 3. Normalized SRTT response time across performance assessment timepoints.** Values less than one (black dashed line) indicate faster than baseline performance. Open circles represent 50 random button presses, and closed circles represent 180 sequenced button presses. One test block occurred in the morning (BL Test) to assess baseline motor performance. The evening session consisted of five training blocks (Training) with test blocks before (PRE Test) and after (POST Test). Error bars represent standard error.

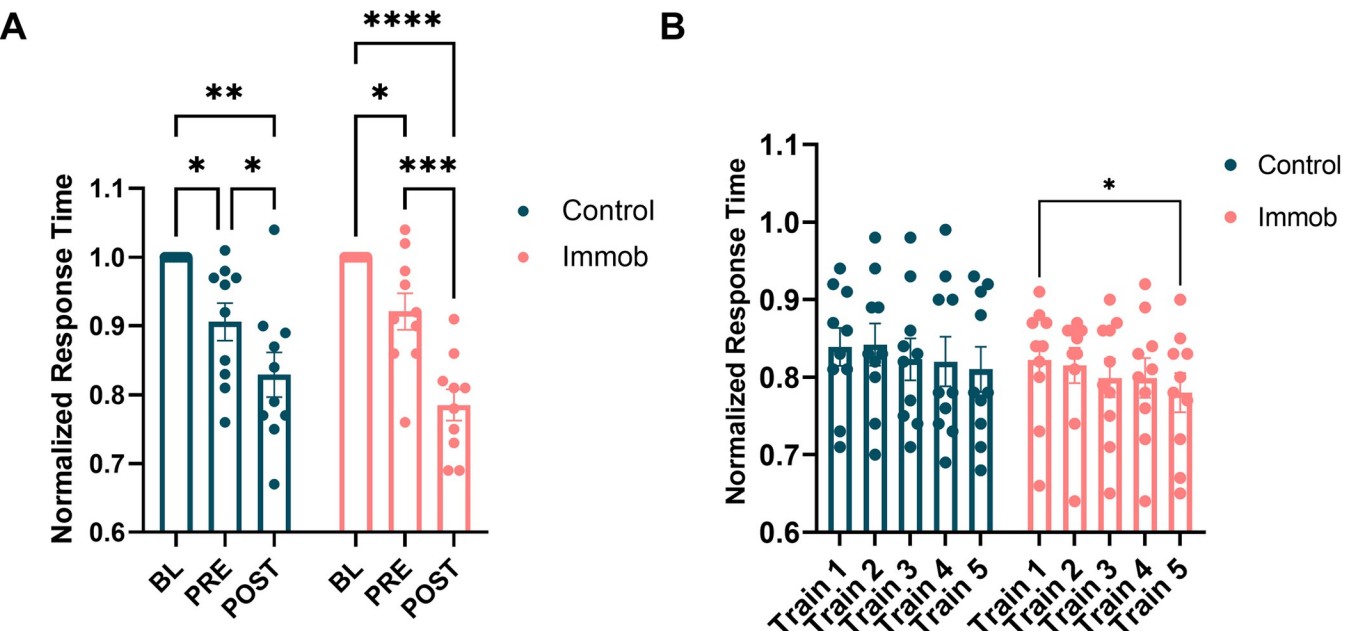

**Fig 4. General motor performance increased with task exposure in both groups.** Normalized response time significantly decreased across training regardless of immobilization condition (F = 57.5, p < .0001). *p < .05, **p < .01, ***p < .001, ****p < .0001.

Random sequence performance increased across test blocks, as measured by a decrease in response time for the last 50 random button presses of each test block, for both control and immobilized participants. A two-way ANOVA indicated that there was a significant main effect of time (F = 57.5, p < .0001) (**Fig 4A**) on normalized response time; however, there was not a main effect of group nor a time X group interaction.

A two-way ANOVA showed a significant main effect of time (F = 6.3, p = .0008) on normalized response time for repeated sequence performance across training blocks 1–5 (**Fig 4B**). There was no effect of group nor a time X group interaction on response time throughout training blocks.

### 3.2 Assessment of sequence-specific skill

Skill score was demonstrated to be normally distributed in the PRE (W = .97, p = .68) and POST (W = .92, p = .11) blocks but not at baseline (W = .83, p = .002). Further assessment using descriptive statistics did not show substantial violations of the assumptions of normality or homogeneity of variance. Interference score was normally distributed at BL (W = .97, p = .85), PRE (W = .96, p = .54), and POST (W = .98, p = .86) timepoints.

Both groups showed an average increase from $SS_{BL}$ to $SS_{PRE}$, but a two-way ANOVA showed no effect of group or time on skill score and no time X group interaction (**Fig 5A**) across all three timepoints. Similarly, a two-way ANOVA showed no effect of group or time nor a time X group interaction on interference scores. **Fig 5B** shows the interference scores for the transition from repeated to random for BL, PRE, and POST test blocks. **Table 2** summarizes the results of each ANOVA for the primary SRTT-based outcome measures.

### 3.3 Task performance of the non-immobilized hand

Response times for the right, non-immobilized hand were determined to be normally distributed in both BL (W = .96, p = .47) and PRE (W = .94, p = .25) blocks. A two-way ANOVA

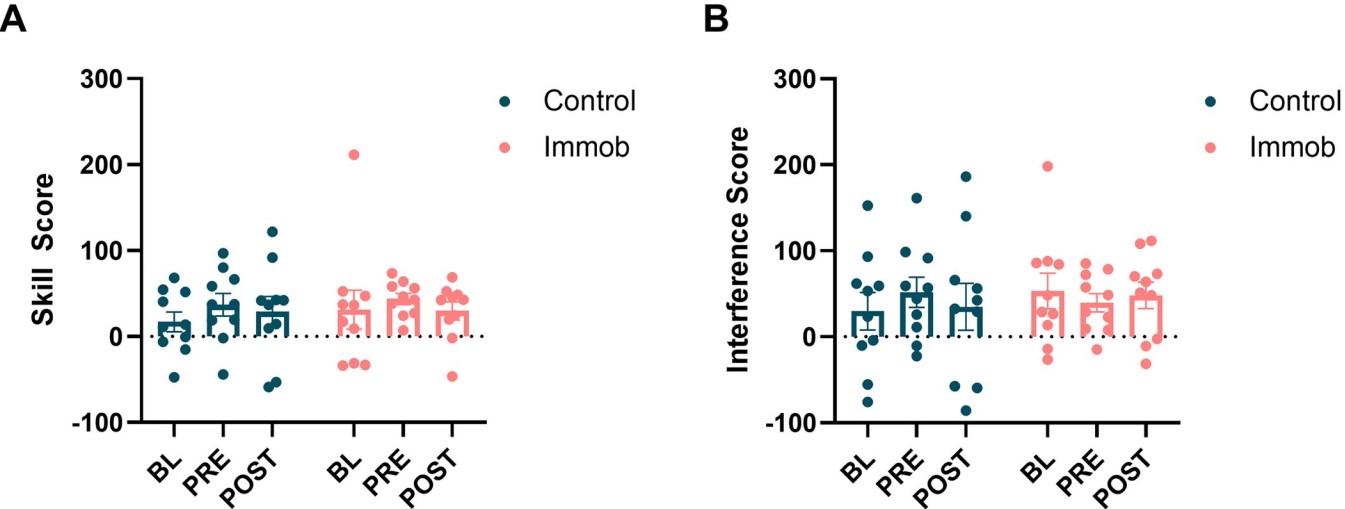

**Fig 5. Sequence-specific skill did not change after training and there was not evidence of an effect of immobilization.** Neither (A) Skill Score nor (B) Interference score significantly changed across task exposure for either group.

demonstrated that normalized response time decreased from BL to PRE timepoints, with a main effect of time (F = 103.2, p < .0001). There was no effect of group, nor a time x group interaction (**Fig 6A**).

Skill score was normally distributed for BL (W = .97, p = .74) and PRE (W = .96, p = .49) blocks. A two-way ANOVA showed no effect of time nor group on skill score, nor a time x group interaction (**Fig 6B**). Full ANOVA results can be found in **Table 3**.

## 4. Discussion

In the current study, we investigated the effect of short-term (6 hours) arm immobilization on implicit motor sequence acquisition. General motor performance improved with task exposure in both groups; however, improvement was not sequence-specific. Despite confirming that the immobilization protocol was followed by individuals in the immobilization group, no

**Table 2. SRTT ANOVA results.**

| Measure | Source | DFn, DFd | F Statistic | p-value | Effect size (eta$^2$) |
|---|---|---|---|---|---|
| Random Sequence Performance | Time | 1.8, 32.6 | 57.5 | **< .0001** | .570 |
| | Group | 1, 18 | 0.16 | .69 | .002 |
| | Time X Group | 2, 36 | 1.4 | .25 | .014 |
| Repeated Sequence Performance | Time | 3.1, 56.5 | 6.3 | **.0008** | .026 |
| | Group | 1, 18 | 0.44 | .51 | .022 |
| | Time X Group | 4, 72 | 0.19 | .95 | .0007 |
| Skill Score | Time | 1.6, 29.6 | 0.73 | .46 | .023 |
| | Group | 1, 18 | 0.30 | .59 | .007 |
| | Time X Group | 2, 36 | 0.12 | .89 | .0034 |
| Interference Score | Time | 1.3, 23.3 | 0.03 | .91 | .001 |
| | Group | 1, 18 | 0.24 | .63 | .005 |
| | Time X Group | 2, 36 | 0.46 | .64 | .016 |

Significant p-values are bolded.

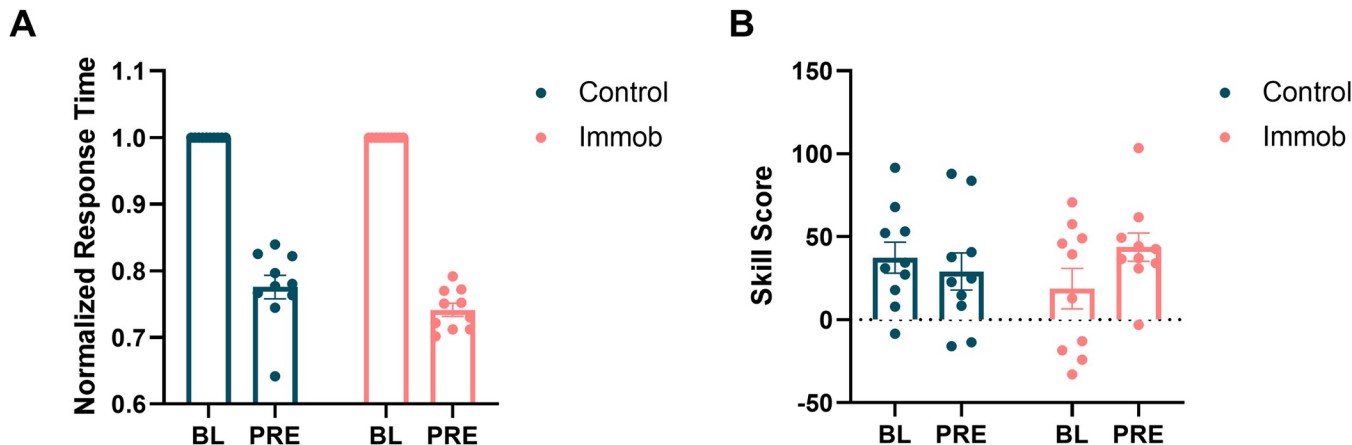

**Fig 6. Immobilization did not significantly influence performance of the right, non-immobilized hand.** (A) Response time decreased from BL to PRE timepoints across participants (F = 103.2, p < .0001), but was not immobilization-specific. (B) Skill score did not significantly change from BL to PRE timepoints across groups.

group differences were found in general motor performance or sequence-specific skill. Overall, our results suggest that immobilization did not significantly augment implicit, sequence-specific skill acquisition or online improvements in general motor performance on the SRTT.

## 4.1 Lack of an effect of immobilization on motor performance on an individuated finger sequencing task

Contrary to the results of previous studies utilizing short-term immobilization [16, 17, 19–24], we did not observe a decrement in motor performance after a period of upper limb immobilization. In fact, general motor performance increased from BL to PRE test blocks in both groups. One potential explanation for the difference in our results could be the duration of immobilization. While changes in measures of corticospinal excitability have been shown in as little as 3 hours after onset of immobilization [16], most previous studies that found impaired motor performance after immobilization used immobilization periods of at least 8 hours. In the current study, arm immobilization occurred for a period of six hours. The idea that duration of immobilization could influence motor performance is supported by a study by Moisello and Bove, which found that motor performance on an out-and-back cursor task decreased after 12, but not 6 hours, of immobilization [21]. Therefore, it is possible that a greater duration of immobilization may be necessary to affect motor performance. Alternatively, limb immobilization may have a greater behavioral effect on tasks that require multi-joint

**Table 3. Nontarget hand motor performance.**

| Measure | Source | DFn, DFd | F Statistic | p-value | Effect size (eta$^2$) |
|---|---|---|---|---|---|
| Random Sequence Performance | Time | 1, 18 | 574.0 | **< .0001** | .93 |
| | Group | 1, 18 | 2.9 | .11 | .005 |
| | Time X Group | 1, 18 | 2.9 | .11 | .005 |
| Skill Score | Time | 1, 18 | .67 | .42 | .016 |
| | Group | 1, 18 | .04 | .85 | .001 |
| | Time X Group | 1, 18 | 2.7 | .12 | .066 |

Significant p-values are bolded.

coordination of the entire limb (e.g., skilled reaching [17]) rather than individual digits in the hand required to perform the SRTT in the current study.

## 4.2 Acquisition of sequential, individuated finger movements was not preferentially enhanced after immobilization

Despite the potential for short-term immobilization to enhance experience-dependent plasticity within M1, our results demonstrated that implicit motor skill acquisition on a task that requires coordination of sequential movements was not influenced by immobilization. While M1 has been shown to be involved in the execution of sequential movements [34, 35], it is unclear whether M1 has a role in acquisition of the sequenced motor skill itself [36]. Therefore, increasing the capacity for LTP-like plasticity in M1 may not be sufficient to influence acquisition of an implicit, sequenced motor skill. Outside of M1, short-term immobilization has been shown to influence markers of plasticity in primary somatosensory cortex [17], but it is unclear whether immobilization influences plasticity in other brain areas that may be involved in acquisition of sequence-specific skill, such as premotor cortex or supplementary motor area [36].

## 4.3 Immobilization of the non-dominant limb did not influence dominant limb motor performance

Even though short-term immobilization has been shown to influence interhemispheric interactions [33], our results demonstrated no evidence of an effect of six hours of immobilization on motor performance of the non-immobilized right hand. Results from previous studies have demonstrated improved performance of the non-immobilized hand on motor tasks after longer periods of immobilization, such as one [37] or two weeks [38]. Behavioral effects of immobilization may differ based on the duration, as longer durations of immobilization (days to weeks) have been shown to lead to additional changes in spinal excitability [39, 40] and cortical morphology [38] that have not been previously observed with shorter durations (6 hours) of immobilization. It is possible that effects of immobilization on the motor performance of the non-immobilized limb is also dependent on the duration of immobilization, and we would predict a similar finding of increased performance of the non-immobilized hand with longer periods of immobilization.

## 4.4 Task characteristics may influence the effects of immobilization

One unique aspect of the current study was that this was the first study to examine the effect of immobilization on a sequence learning task that relies on individuated finger movements. Therefore, the different characteristics of the task itself as well as the outcome measures could have contributed to inconsistent results between this study and others. Several previous studies that observed a decrease in motor performance after a period of immobilization used tasks that required control of more proximal portions of the upper extremity, such as reaching, while the current study used a modified SRTT that emphasized fine control of the distal upper extremity. Previous research has demonstrated that the composition of descending projections to the distal and proximal upper extremity are different in primates [41–43], and it is possible that immobilization differentially modulates these pathways. Bolzoni and Bruttini suggested that function of the proximal muscles responsible for postural control is more likely to be influenced by a period of immobilization, even when only distal hand muscles are immobilized [20]. Similarly, in a task requiring participants to pick up and put down a pencil repeatedly, immobilization of the dominant arm increased reach duration and changed acceleration and

deceleration of movement but did not influence grip aperture [19]. In the current study, outcome measures to assess general motor performance and sequence-specific skill were calculated using response time, which are not able detect changes in joint kinematics that may occur after a period of immobilization. Future studies can quantify joint coordination during task performance with kinematic data [44] and/or separating response time into reaction time and movement time to examine central nervous system contributions to changes in SRTT performance with training and immobilization [45].

Another potential explanation for the observed findings is that immobilization impairs proprioceptive processing, and tasks that require proprioceptive information to complete will be impacted by a period of immobilization. This idea is supported by Avanzino and Pelosin that showed that neurophysiological changes normally seen after immobilization were blocked when proprioceptive receptors were selectively activated during the immobilization period [46]. This could explain why performance on the modified SRTT, which required small amplitude, individuated finger movements that would not be expected to be affected by postural control or modulation of proprioceptive receptor activity, was not negatively impacted by immobilization. Taken together, our findings support prior literature suggesting that multi-joint coordination of arm movements may be preferentially impacted by upper limb immobilization. It remains unclear if immobilization can modulate the acquisition of skill for tasks requiring multi-joint coordination of the arm.

### 4.5 Skill improvements across groups were not sequence-specific

An unexpected result from the current study was that sequence-specific skill did not significantly improve after training. One possible explanation for the lack of change in skill score and interference score in across training is that time of day influenced sequence-specific skill acquisition, since all motor training was performed in the evening session for the current group of participants in our study. Previous research has suggested that skill improvement on a sequence learning task is greater in the morning compared to the evening [47], which does not seem to be the case with acquisition of skill in a repetitive ballistic motor training task [48]. Interestingly, Keisler and Ashe suggested that motor sequence learning itself may not be impaired in the evening relative to morning, but factors such as motivation, attention, and fatigue may lead to the impairment of the expression of learning (in the form of task performance) [49]. Additionally, previous research has shown that performance during a skill acquisition task cannot be equated with skill learning. In fact, certain features of a motor task itself, such as task difficulty or practice structure, can lead to a decrease in performance during the acquisition phase of learning but subsequently enhance retention of skill during a follow-up assessment [50, 51]. Including delayed retention testing could assess the effect of immobilization on motor sequence learning when training occurs in the evening.

### 4.6 Study limitations

There are several limitations to the current study. The current study only assessed within-session skill acquisition, thus, the effect of upper limb immobilization on skill learning remains unknown. Skill retention and generalization can be evaluated in future studies to determine if immobilization has an effect on skill learning. A priori sample size calculations were based on pilot work and previously published studies showing large effect sizes of immobilization on motor performance. Although the effect of immobilization on motor skill acquisition in the current study was consistent with the hypothesized direction, the observed effects sizes were small. Future studies with larger sample sizes can be conducted to detect small effect sizes, if present, or to test equivalence. Additionally, it is possible that the color-matching component

of the task may have masked the sequence, making sequence-specific acquisition, even implicitly, more difficult that could have contributed to the lack of change in sequence-specific skill after a single training session. Future studies could employ different versions of the SRTT to address this limitation or investigate other tasks more closely aligned with the effects of arm immobilization (e.g., skilled reaching movements).

## 5. Conclusions

Overall, our results suggest that short-term (6 hours) immobilization of the arm has a small effect on implicit skill acquisition on a task that requires individuated, sequenced finger movements. However, it is possible that task characteristics and the duration of immobilization influenced the results. These initial findings suggest the behavioral effects of short-term arm immobilization may be task specific and depend on duration of immobilization. Future studies should assess the effects of immobilization on skill acquisition and learning using tasks that require multi-joint control and/or proprioceptive feedback to understand the capacity for immobilization to augment endogenous experience-dependent plasticity associated with training or task-specific rehabilitation.

## Acknowledgments

The authors would like to acknowledge Scott Heston for his contributions to programming the motor task and creating the button boxes used during the task, as well as Maria Krakovski and Martin Tan for their assistance with data collection and processing.

## Author Contributions

**Conceptualization:** Erin M. King, Lauren L. Edwards, Michael R. Borich.

**Data curation:** Erin M. King.

**Formal analysis:** Erin M. King, Michael R. Borich.

**Funding acquisition:** Erin M. King.

**Investigation:** Erin M. King, Michael R. Borich.

**Methodology:** Erin M. King, Michael R. Borich.

**Project administration:** Erin M. King, Michael R. Borich.

**Resources:** Michael R. Borich.

**Software:** Michael R. Borich.

**Supervision:** Michael R. Borich.

**Validation:** Michael R. Borich.

**Visualization:** Erin M. King, Michael R. Borich.

**Writing – original draft:** Erin M. King.

**Writing – review & editing:** Erin M. King, Lauren L. Edwards, Michael R. Borich.

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
