## [Decision Letter · Decision Letter 0]

20 Sep 2021

PONE-D-21-19679Effects of short-term arm immobilization on motor skill acquisitionPLOS ONE

Dear Dr. Borich,

Thank you for submitting your manuscript to PLOS ONE. After careful consideration, we feel that it has merit but does not fully meet PLOS ONE’s publication criteria as it currently stands. Therefore, we invite you to submit a revised version of the manuscript that addresses the points raised during the review process. Please make sure you consider the concerns of Reviewer 1 about whether a TMS experiment was performed (and not reported) in these participants. To limit publication bias in the literature (and appropriate multiple-comparison correction), it is critical to report negative results rather than omitting them. If additional TMS hypotheses were tested but not reported, the manuscript may require substantial rewriting. If not, then the manuscript needs to justify why TMS exclusion/timing criteria were used.

We look forward to receiving your revised manuscript.

Kind regards,

Benjamin A. Philip

Academic Editor

PLOS ONE

“The authors would like to acknowledge Scott Heston for his contributions to programming the motor task and creating the button boxes used during the task, as well as Maria Krakovski and Martin Tan for their assistance with data collection and processing. The authors would also like to acknowledge their funding sources. EMK is supported by NIH 2T32HD071845-06.”

“EMK is supported by the Emory Mechanisms of Learning Across Development and Species Training Grant 2T32HD071845-06 (https://sites.google.com/view/mechanismsoflearning/home), a Ruth L. Kirschstein Institutional National Research Service Award through the National Institutes of Health. The funders had no role in study design, data collection and analysis, decision to publish, or preparation of the manuscript.”

Additional Editor Comments (if provided):

Reviewers' comments:

Reviewer's Responses to Questions

**Comments to the Author**

1. Is the manuscript technically sound, and do the data support the conclusions?

Reviewer #1: Partly

Reviewer #2: Yes

2. Has the statistical analysis been performed appropriately and rigorously? 

Reviewer #1: No

Reviewer #2: Yes

3. Have the authors made all data underlying the findings in their manuscript fully available?

Reviewer #1: Yes

Reviewer #2: Yes

4. Is the manuscript presented in an intelligible fashion and written in standard English?

Reviewer #1: Yes

Reviewer #2: Yes

5. Review Comments to the Author

Reviewer #1: In the current study, the authors investigated the influence of plasticity mechanisms via limb immobilization on implicit skill acquisition. This work is theoretically driven and generally well written. Both control and immobilized groups improved the random and repeated sequences, without any difference between the groups.

While this topic is of interest, I have general and specific concerns that limit the value of the current findings.

General remarks:

- Even if it is clear and well written, the introduction focuses a lot on neurophysiogical aspects of immobilization. Considering, however, this research field as well as the used experimental paradigm, I personally think that the introduction section lacks of a paragraph that focuses on behavioral results. To present behavioral literature at the end of the introduction would give a clearer overview of the state of the art and to build a bridge with the method section.

- Also, given the introduction, as well as the lines 79-80, one would expect neurophysiological measurements to support the behavioral predictions (e.g., to check if corticospinal excitability decreased after immobilization). Also, if there is no method nor results for transcranial magnetic stimulation, why is the contraindication to TMS an exclusion criterion? Unfortunately, this could suggest that TMS has been used, but that the results were not reported, raising question about the reasons that could have motivated such decision.

- The main conclusion is that short-term arm immobilization does not modulate motor acquisition in SRTT. First, and as stated by the authors themselves, quite comparable results have been also observed in Moisello et al. (2011). It seems that a minimum of 8-hour immobilization is required to induce behavioral changes. Could the authors clarify the rationale for the 6-hour immobilization? Also, based on the current analysis, the claim that short-term immobilization does not modulate motor acquisition, and that “Acquisition of sequential, individuated finger movements is not preferentially enhanced” cannot be stated in such way. One cannot conclude on the absence on an effect with such statistics. For that purpose, I recommend to perform specific statistics, such as equivalence testings (see Lakens et al., 2018).

- Could the author clarify the experimental design? What was the purpose of performing the repeated sequence between 2 blocks of random sequences? What is the goal of the second block of random sequence?

- Could the authors specify in the abstract and/or the conclusion, when missing, the duration of the immobilization (which seems to be important to induce behavioral changes) and that it was an implicit sequence task?

Specific comments:

Abstract

Line 25: I would suggest to remove “younger”, or to replace it by “young”.

Methods

Lines 136-137: Could you please provide information about which data proportion have been removed? And were trials with RT below 100ms removed?

Line 142: Has the normality been verified prior to the analyses?

Lines 151: Sequence-specific skill.

- Skill score: 48 button presses of repeated sequence (at baseline, pre and post) minus the 50 button presses of the last random sequence at baseline. Why did the authors choose the last random sequence at baseline?

Line 170: Please provide here information about which post-hocs were used.

Results

Please provide effect sizes and the full degrees of freedom for each reported Anovas.

It may be a personal misunderstanding, but I do not understand why the post hocs are reported with the letters "t" or "q".

l.197: results of random sequence performance. The authors did not show any statistical results for this outcome in the text. It seems we have to wait for Table 1 to see the results. Please add information in the text for clarity. How is the statistical design for the random sequences? Did the authors merge the first and the second block of each test? Or was it integrated as a specific factor within the ANOVA?

l.201-205: As there was only a main effect of Time, it is not relevant to show separate comparisons for each group.

l.213: As there was only a main effect of Time for the training data analysis, it is not accurate to perform post-hoc tests for each group individually (“the normalized response time for training block 5 was significantly faster than training block 1 in the immobilized participants (t=4.2, p=.024) but not the control participants (t=2.5, p=.30)”).

About the results of the training block, how did the authors normalize data of the repeated sequences? In Figure 2 (with all the blocks and sequences), it seems that 1) RT of the last 3 blocks of the training were shorter than RT at post for the repeated sequence and 2) RT for the immobilized group are shorter, especially at block 5, in comparison to the control group.

Discussion

The authors should revise the discussion based on the appropriate statistics.

Figure and Tables

Figures 2, 3 and 4 are intermixed in the text.

l.179: “Fig 2. Immobilized participants complied with the immobilization procedure”.

l.187: “Normalized response times for each group across training are shown in Figure 3”.

In Table 1, for the repeated sequence performance, what data are presented? The raw data or the normalized data?

Reviewer #2: King and colleagues investigated whether short-term arm immobilization influences motor skill acquisition by employing a modified Serial Reaction Time Task (SRTT) in healthy adults. The motor skill acquisition was assessed after six hours of immobilization of the non-dominant arm (experimental group) or an equivalent period of no immobilization (control group). Results showed an improved performance overall after training, not influenced by immobilization. The authors conclude that six-hour immobilization does not augment motor skill acquisition for sequential, individuated finger movements.

The topic addressed in this paper is relevant. The manuscript is overall nicely written and easy to follow. However, before publication, I have a few general suggestions that might be worth considering in the introduction and discussion and a few minor suggestions regarding the methods.

Introduction

The introduction contains a detailed description of the studies that have investigated the effect of limb immobilization on synaptic processes, long-term potentiation, and long-term depression. Whereas this background can give a broad idea of limb immobilization, it does not provide the rationale for this study. It is not clear why we should study short-term limb immobilization for motor skill training-induced plasticity, since it has already largely been investigated with different tasks (e.g., Moisello et al., 2008; Ngomo et al., 2012 https://doi.org/10.1016/j.neuroscience.2012.06.018; Opie et al., 2016; Lundbye Jensen et al, 2005 https://doi.org/10.1152/japplphysiol.01408.2004). What is the novelty of the present study?

Regarding limb immobilization, the authors only comment on studies that investigated the motor system with non-invasive brain stimulation techniques. However, it could be interesting to read a few words about the M1 activity after immobilization with different other methods (e.g., Huber et al. 2006 https://doi.org/10.1038/nn1758; Avanzino et al. 2011 https://doi.org/10.1523/JNEUROSCI.4893-10.2011; Garbarini et al., 2018 doi: 10.1093/cercor/bhy134; Langer et al. 2012 doi: 10.1212/WNL.0b013e31823fcd9c; and, in a similar vein Sperl et al., 2021 https://doi.org/10.1007/s00221-021-06190-w).

The authors predict that short-term limb immobilization would lead to a greater motor skill acquisition with training that followed immobilization, based on increased capacity for activity-dependent synaptic strengthening in the corresponding contralateral M1 representation. However, the opposite prediction can be hypothesized since studies found that even short limb immobilization affects motor performance (e.g., Moisello et al., 2008).

Methods

This study has a between-subjects design, with ten participants per group. In my opinion, this is a minimal sample size, is there a reason? If an a priori analysis about sample size was not performed, I think the authors should add this as a limitation of the study.

Another consideration is that probably an interesting control for this task should have been the no-immobilized hand, instead of a control group. In this way, the authors could have used a within-subjects design. What do expect to find if they had used the dominant hand as a control, in light of the literature about use-dependent hemispheric balance (see for example Avanzino et al., 2011 https://doi.org/10.1523/JNEUROSCI.4893-10.2011)?

Did the authors collect accuracy of the responses too?

Discussion

In a recent study on motor inhibition and limb immobilization (Bruno et al., 2020 https://doi.org/10.1016/j.neuroimage.2020.116911), the authors found no modulation of reaction times (RTs) in a Go/Nogo task between pre- and post immobilization (one week) of the non dominant hand. On the contrary, they found a better performance (lower RTs) for the dominant no-immobilized hand. Is this result in line with the findings of the present manuscript? Can di author comment on this?

The result about the better performance (lower RTs in train 5 as compared to train 1, figure 4B) in the immobilized group is not discussed. Can the authors spend a few words on this? In some way, this is a result suggesting that the immobilization induced a better motor skill.

Minor

Why the TMS requirements had to be respected? In this study, TMS was not employed.

Why did the authors choose a six-hour immobilization?

The name of the figures is wrong (figure 2 is called 3, and vice versa).

I did not find in the text the link with the raw data.

6. PLOS authors have the option to publish the peer review history of their article (what does this mean?). If published, this will include your full peer review and any attached files.

Reviewer #1: No

Reviewer #2: No

---

## [Author Response · Author response to Decision Letter 0]

17 Mar 2022

Response to Reviewers

Reviewer #1: In the current study, the authors investigated the influence of plasticity mechanisms via limb immobilization on implicit skill acquisition. This work is theoretically driven and generally well written. Both control and immobilized groups improved the random and repeated sequences, without any difference between the groups.

While this topic is of interest, I have general and specific concerns that limit the value of the current findings.

Thank you for the positive and constructive comments regarding the original submission. We have extensively revised the manuscript based on the concerns raised to increase the value of the observed findings to the readership of PloS ONE.

General remarks:

- Even if it is clear and well written, the introduction focuses a lot on neurophysiogical aspects of immobilization. Considering, however, this research field as well as the used experimental paradigm, I personally think that the introduction section lacks a paragraph that focuses on behavioral results. To present behavioral literature at the end of the introduction would give a clearer overview of the state of the art and to build a bridge with the method section.

We appreciate this comment that increases the connection between the current project objective and the behavioral literature. Additional information has been added to the Introduction section (lines 70-77) to contextualize the current study with existing behavioral literature. 

- Also, given the introduction, as well as the lines 79-80, one would expect neurophysiological measurements to support the behavioral predictions (e.g., to check if corticospinal excitability decreased after immobilization). Also, if there is no method nor results for transcranial magnetic stimulation, why is the contraindication to TMS an exclusion criterion? Unfortunately, this could suggest that TMS has been used, but that the results were not reported, raising question about the reasons that could have motivated such decision.

We appreciate identifying this oversight. This inclusion criterion was part of a separate experiment that included TMS delivery, and it has been removed from the revised submission for clarity. 

- The main conclusion is that short-term arm immobilization does not modulate motor acquisition in SRTT. First, and as stated by the authors themselves, quite comparable results have been also observed in Moisello et al. (2011). It seems that a minimum of 8-hour immobilization is required to induce behavioral changes. Could the authors clarify the rationale for the 6-hour immobilization? Also, based on the current analysis, the claim that short-term immobilization does not modulate motor acquisition, and that “Acquisition of sequential, individuated finger movements is not preferentially enhanced” cannot be stated in such way. One cannot conclude on the absence on an effect with such statistics. For that purpose, I recommend to perform specific statistics, such as equivalence testings (see Lakens et al., 2018).

Thank you for raising these important points. We selected a six-hour immobilization period both to increase feasibility for the experimental protocol based on findings of our pilot work and to support the likelihood of future translation to real-world and/or clinical settings. Given the literature, it is possible that at least 8hrs of immobilization is necessary for a behavioral effect. We have expanded on this point in the revised Discussion section (lines 379-383, 413-416).

We agree that it is not possible to determine the absence of an effect with the statistical analyses performed. The experimental design and a priori sample size calculation was not intended for formal equivalence/non-inferiority testing. We have softened the language in the Discussion section to clarify the interpretation of observed findings that there was a lack of evidence of an effect of six hours of immobilization on motor skill acquisition. 

We have also revised the Limitations section in the Discussion to clarify that the findings observed were in the hypothesized direction, but the effect sizes were smaller than previous studies used to power the present study and the study was not powered to determine equivalence or to detect small effect sizes. Future studies with larger sample sizes could be performed to build upon the current findings (lines 489-497).

- Could the author clarify the experimental design? What was the purpose of performing the repeated sequence between 2 blocks of random sequences? What is the goal of the second block of random sequence?

In line with previous studies utilizing a similar modified version of the SRTT (Robertson et al., 2004; Cohen et al., 2005), the test block has a “sandwich” design, consisting of repeated sequence button presses flanked on either side by random button presses. The skill score is derived from the last 48 repeated button presses (4 repeats of the 12-item sequence) and the second block of random sequence button presses. Having random trials before and after the block of repeated sequences at each time point provided a balanced test block design and offered the opportunity to compare our results to previous studies utilizing the SRTT. We have added references to the Methods (lines 107, 175-176)

- Could the authors specify in the abstract and/or the conclusion, when missing, the duration of the immobilization (which seems to be important to induce behavioral changes) and that it was an implicit sequence task?

Thank you for this request. The information has been added throughout. 

Specific comments:

Abstract

Line 25: I would suggest to remove “younger”, or to replace it by “young”.

Thank you for noting this language which we usually use when evaluating group differences between ‘younger’ and ‘older’ adult cohorts. In the current study, we have revised to ‘young’ as suggested.

Methods

Lines 136-137: Could you please provide information about which data proportion have been removed? And were trials with RT below 100ms removed?

Overall, 1.99% of button presses were removed (694/34,800 total button presses) across all participants (range: 1.38%-2.76%). This information has been added to the text (lines 234-235). No response times were below 100ms, although that was not a criterion for removal. It was possible that implicit knowledge of the sequence could allow for the participants to press the correct button before processing the visual stimulus; however, evidence was not found for this effect in the current study.

Line 142: Has the normality been verified prior to the analyses?

Thank you for noting this point missing from the original manuscript. Normality was statistically assessed with the Shapiro-Wilk's test and inspection of data distribution and Q-Q plots to verify homogeneity of variance. This information has been added to the text in the Methods and Results sections.

Lines 151: Sequence-specific skill.

- Skill score: 48 button presses of repeated sequence (at baseline, pre and post) minus the 50 button presses of the last random sequence at baseline. Why did the authors choose the last random sequence at baseline?

In line with previous studies using a similar version of the SRTT and task design (Robertson et al., 2004; Cohen et al., 2005), the skill score was calculated as the response time for the last 50 random button presses of each test block minus the 48 button presses of repeated sequence immediately preceding it. We have included this information in the text to make it clearer. The last random and repeated sequences were used in all test blocks in order to account for possible within-test block order effects. 

Line 170: Please provide here information about which post-hocs were used.

Sidak’s multiple comparisons tests were used when a time x group interaction was found. This information has been added to the Results section (line 216).

Results

Please provide effect sizes and the full degrees of freedom for each reported Anovas.

Effect sizes (eta squared) and degrees of freedom have been added for all ANOVA tables. 

It may be a personal misunderstanding, but I do not understand why the post hocs are reported with the letters "t" or "q".

Post-hoc t-tests have been removed (see below comments for context).

l.197: results of random sequence performance. The authors did not show any statistical results for this outcome in the text. It seems we have to wait for Table 1 to see the results. Please add information in the text for clarity. How is the statistical design for the random sequences? Did the authors merge the first and the second block of each test? Or was it integrated as a specific factor within the ANOVA?

This information has been included and minor edits have been made for clarity. We only included the second set of random button presses in each test block in the analysis for general motor performance. We performed this analysis to evaluate general improvements in motor performance on the SRTT associated with both immobilization and training on the repeated sequences.

l.201-205: As there was only a main effect of Time, it is not relevant to show separate comparisons for each group.

Considering this comment, we have removed the separate comparisons for each group.

l.213: As there was only a main effect of Time for the training data analysis, it is not accurate to perform post-hoc tests for each group individually (“the normalized response time for training block 5 was significantly faster than training block 1 in the immobilized participants (t=4.2, p=.024) but not the control participants (t=2.5, p=.30)”).

The post-hoc tests have been removed. 

About the results of the training block, how did the authors normalize data of the repeated sequences? In Figure 2 (with all the blocks and sequences), it seems that 1) RT of the last 3 blocks of the training were shorter than RT at post for the repeated sequence and 2) RT for the immobilized group are shorter, especially at block 5, in comparison to the control group.

Average response times were all normalized to the first 50 random button presses in the Baseline test block. Response times do appear to have qualitatively increased from the end of training to the post-training block, which could have been due to the transition from repeated button presses to random button presses. Other alternatives include the short break (30 seconds to a minute) between the last training block and the post-test block leading to an increase in response time, or there could be an effect of fatigue. Mean RT was shorter for the immobilized group towards the end of training, but due to interindividual variability, the difference did not reach statistical significance. 

Discussion

The authors should revise the discussion based on the appropriate statistics.

We have made several revisions to the Discussion section to more closely align with the results from the updated statistical analyses performed. 

Figure and Tables

Figures 2, 3 and 4 are intermixed in the text.

l.179: “Fig 2. Immobilized participants complied with the immobilization procedure”.

l.187: “Normalized response times for each group across training are shown in Figure 3”.

The figure names have been updated to correct this issue identified 

In Table 1, for the repeated sequence performance, what data are presented? The raw data or the normalized data?

The normalized data are provided in Table 1. 

Reviewer #2: King and colleagues investigated whether short-term arm immobilization influences motor skill acquisition by employing a modified Serial Reaction Time Task (SRTT) in healthy adults. The motor skill acquisition was assessed after six hours of immobilization of the non-dominant arm (experimental group) or an equivalent period of no immobilization (control group). Results showed an improved performance overall after training, not influenced by immobilization. The authors conclude that six-hour immobilization does not augment motor skill acquisition for sequential, individuated finger movements.

The topic addressed in this paper is relevant. The manuscript is overall nicely written and easy to follow. However, before publication, I have a few general suggestions that might be worth considering in the introduction and discussion and a few minor suggestions regarding the methods.

Thank you for the constructive review and positive remarks regarding the original submission. We have incorporated each suggestion into the revised manuscript to enhance the clarity and utility of the project findings

Introduction

The introduction contains a detailed description of the studies that have investigated the effect of limb immobilization on synaptic processes, long-term potentiation, and long-term depression. Whereas this background can give a broad idea of limb immobilization, it does not provide the rationale for this study. It is not clear why we should study short-term limb immobilization for motor skill training-induced plasticity, since it has already largely been investigated with different tasks (e.g., Moisello et al., 2008; Ngomo et al., 2012 https://doi.org/10.1016/j.neuroscience.2012.06.018; Opie et al., 2016; Lundbye Jensen et al, 2005 https://doi.org/10.1152/japplphysiol.01408.2004). What is the novelty of the present study?

Several studies have assessed the effect of immobilization on motor performance, but few have assessed the effect of immobilization on skill acquisition, and none have assessed the effect of immobilization on acquisition of skill requiring sequenced, individuated finger movements. Some context regarding how the current study fits in with existing studies of motor performance after immobilization has been added to the Introduction section (lines 70-77).

Regarding limb immobilization, the authors only comment on studies that investigated the motor system with non-invasive brain stimulation techniques. However, it could be interesting to read a few words about the M1 activity after immobilization with different other methods (e.g., Huber et al. 2006 https://doi.org/10.1038/nn1758; Avanzino et al. 2011 https://doi.org/10.1523/JNEUROSCI.4893-10.2011; Garbarini et al., 2018 doi: 10.1093/cercor/bhy134; Langer et al. 2012 doi: 10.1212/WNL.0b013e31823fcd9c; and, in a similar vein Sperl et al., 2021 https://doi.org/10.1007/s00221-021-06190-w).

Thank you for this comment and opportunity to include additional relevant findings investigating M1 activity post-immobilization. We have added description of neurophysiological changes associated with immobilization using methods other than TMS in the Introduction (line 65) and Discussion (lines 407-409). 

The authors predict that short-term limb immobilization would lead to a greater motor skill acquisition with training that followed immobilization, based on increased capacity for activity-dependent synaptic strengthening in the corresponding contralateral M1 representation. However, the opposite prediction can be hypothesized since studies found that even short limb immobilization affects motor performance (e.g., Moisello et al., 2008).

We agree that immobilization has been shown to affect performance of certain types of motor tasks including skilled reaching. Based on the previous literature demonstrating reduced motor performance after immobilization, we hypothesized that the ability to perform a serial key press task would be temporarily reduced after immobilization, but the ability to acquire sequence-specific motor skill would be enhanced based the concept of increasing the dynamic range of synaptic strengthening through immobilization. Investigating both performance on random and repeated sequences offered the opportunity to concomitantly evaluate the effects of immobilization on both general motor performance and sequence-specific skill acquisition. Although the observed effect sizes were small, the direction of effect was in line with the hypothesis that immobilization may increase the dynamic range for synaptic strengthening. We have clarified these points in the Introduction and Discussion sections.

Methods

This study has a between-subjects design, with ten participants per group. In my opinion, this is a minimal sample size, is there a reason? If an a priori analysis about sample size was not performed, I think the authors should add this as a limitation of the study.

An a priori sample size was performed based on previous immobilization studies and our pilot work. However, the effect sizes observed in previous published studies were large (e.g., Moisello et al., 2008) in comparison to the small effects observed in the current study. We have acknowledged the limited sample size considering the small effect sizes observed in the current study which has been added as a potential limitation in the Discussion section. 

Another consideration is that probably an interesting control for this task should have been the no-immobilized hand, instead of a control group. In this way, the authors could have used a within-subjects design. What do expect to find if they had used the dominant hand as a control, in light of the literature about use-dependent hemispheric balance (see for example Avanzino et al., 2011 https://doi.org/10.1523/JNEUROSCI.4893-10.2011)?

Thank you raising this important point. SRTT data from the right hand were collected before and after the immobilization period (BL and PRE test timepoints) to assess motor performance of the non-immobilized hand. Based on the results from Avanzino et al., 2011, we would have predicted an increase in performance of the non-immobilized hand after the immobilization period. However, we found no evidence of an effect of six hours of immobilization on the performance of the non-immobilized hand. This could be due to factors such as the duration of immobilization. We have added this analysis and interpretation to the revised manuscript. 

Did the authors collect accuracy of the responses too?

Yes, accuracy data were collected for all participants. Given the relatively low nominal difficulty of the task, accuracy was, as expected, high and was similar for both groups. The overall accuracy for control participants was 97.4%, and the overall accuracy for immobilized participants was 96.4%. This information has been added to the Results section (lines 240-241).

Discussion

In a recent study on motor inhibition and limb immobilization (Bruno et al., 2020 https://doi.org/10.1016/j.neuroimage.2020.116911), the authors found no modulation of reaction times (RTs) in a Go/Nogo task between pre- and post immobilization (one week) of the non dominant hand. On the contrary, they found a better performance (lower RTs) for the dominant no-immobilized hand. Is this result in line with the findings of the present manuscript? Can di author comment on this?

Thank you for raising this interesting point and recent finding. It is difficult to compare studies utilizing immobilization on the scale of hours to studies utilizing immobilization on the scale of days to weeks because an immobilization duration of a week has the potential to elicit larger effects on corticospinal excitability (Clark et al., 2010; Lundbye-Jensen et al., 2008) and even cortical thickness (Langer et al., 2012) that have not been previously shown in shorter (6hr) durations of immobilization. Although we did not observe a significant effect of immobilization on SRTT performance with the non-immobilized hand, we did observe small effects in the hypothesized direct, thus, we would predict a similar finding of increased non-immobilized performance on this task with longer periods of immobilization. This point has been added to the Discussion section (Section 4.3, lines 400-412).

The result about the better performance (lower RTs in train 5 as compared to train 1, figure 4B) in the immobilized group is not discussed. Can the authors spend a few words on this? In some way, this is a result suggesting that the immobilization induced a better motor skill.

Although we did observe small effects in the hypothesized direction, the degree of inter-individual variability in performance change during training detect potential between-group differences in performance within the training period. This information has also been added to the Discussion section (lines 488-493).

Minor

Why the TMS requirements had to be respected? In this study, TMS was not employed.

Thank you for identifying this oversight. The TMS inclusion criterion was part of a separate experiment, and it has been removed from this document for clarity.

Why did the authors choose a six-hour immobilization?

Previous literature has shown TMS markers of cortical plasticity as soon as three hours after the onset of immobilization (Karita et al., 2017). Six hours of immobilization was chosen in order for task training to occur after synaptic plasticity has been induced. Also, six hours compared to 8-12 hours was chosen to reduce participant burden and move towards clinical translation if it were to be used as a potential plasticity-inducing intervention. We have added discussion of this point in the Discussion section. 

The name of the figures is wrong (figure 2 is called 3, and vice versa).

Thank you for identifying this error. We have corrected the figure titles. 

I did not find in the text the link with the raw data.

We have added the link to the raw data within the text of the revised manuscript (line 143-144). The link can also be found here: https://osf.io/wazdy/?view_only=cfed7648613f45589c86d73582f93ce7

---

## [Decision Letter · Decision Letter 1]

18 Apr 2022

PONE-D-21-19679R1Effects of short-term arm immobilization on motor skill acquisitionPLOS ONE

Dear Dr. Borich,

Thank you for submitting your manuscript to PLOS ONE. After careful consideration, we feel that it has merit but does not fully meet PLOS ONE’s publication criteria as it currently stands. Therefore, we invite you to submit a revised version of the manuscript that addresses the points raised during the review process.

This manuscript is nearly ready for publication, but please address Reviewer 2's minor comments #1-2.

We look forward to receiving your revised manuscript.

Kind regards,

Benjamin A. Philip

Academic Editor

PLOS ONE

Journal Requirements:

Reviewers' comments:

Reviewer's Responses to Questions

**Comments to the Author**

1. If the authors have adequately addressed your comments raised in a previous round of review and you feel that this manuscript is now acceptable for publication, you may indicate that here to bypass the “Comments to the Author” section, enter your conflict of interest statement in the “Confidential to Editor” section, and submit your "Accept" recommendation.

Reviewer #1: (No Response)

Reviewer #2: All comments have been addressed

2. Is the manuscript technically sound, and do the data support the conclusions?

Reviewer #1: Yes

Reviewer #2: Yes

3. Has the statistical analysis been performed appropriately and rigorously? 

Reviewer #1: No

Reviewer #2: Yes

4. Have the authors made all data underlying the findings in their manuscript fully available?

Reviewer #1: Yes

Reviewer #2: Yes

5. Is the manuscript presented in an intelligible fashion and written in standard English?

Reviewer #1: Yes

Reviewer #2: Yes

6. Review Comments to the Author

Reviewer #1: The authors responded point-by-point to all comments.

I have only minor comments regarding their responses.

1) To test the homogeneity of variance, the authors should use Levene’s test, not Q-Q plots

2) When reporting the degree of freedom, they should report both the numerator and the denominator

3) The references have to be homogenized (e.g., l.68: A study by Opie, Evans (22)…)

Reviewer #2: The authors have substantially improved their manuscript and addressed all of my concerns. Thank you for taking into account my suggestions.

7. PLOS authors have the option to publish the peer review history of their article (what does this mean?). If published, this will include your full peer review and any attached files.

Reviewer #1: **Yes: **Florent Lebon, PhD

Reviewer #2: No

---

## [Author Response · Author response to Decision Letter 1]

1 Jun 2022

Response to Reviewer Comments:

1) To test the homogeneity of variance, the authors should use Levene’s test, not Q-Q plots

Thank you for this suggestion. We performed Levene’s test to assess homogeneity for each outcome measure across timepoints for both left and right hand data. The variances were not significantly different for any of the outcome measures. This information has been added to the manuscript. 

2) When reporting the degree of freedom, they should report both the numerator and the denominator

Thank you for pointing this issue out. The numerator and denominator of the degrees of freedom (DFn, DFd) have been added to the tables. 

3) The references have to be homogenized (e.g., l.68: A study by Opie, Evans (22)…)

The references have been edited throughout to improve consistency in formatting.

---

## [Editor Report · Decision Letter 2]

17 Jun 2022

PONE-D-21-19679R2Effects of short-term arm immobilization on motor skill acquisitionPLOS ONE

Dear Dr. Borich,

Thank you for submitting your manuscript to PLOS ONE. After careful consideration, we feel that it has merit but does not fully meet PLOS ONE’s publication criteria as it currently stands. Therefore, we invite you to submit a revised version of the manuscript that addresses the points raised during the review process.

 This manuscript is nearly ready for acceptance, but please address the remaining minor concerns by Reviewer 1. 

We look forward to receiving your revised manuscript.

Kind regards,

Benjamin A. Philip

Academic Editor

PLOS ONE
---

## [Author Response · Author response to Decision Letter 2]

2 Sep 2022

1) To test the homogeneity of variance, the authors should use Levene’s test, not Q-Q plots

Thank you for this suggestion. We performed Levene’s test to assess homogeneity for each outcome measure across timepoints for both left and right hand data. The variances were not significantly different for any of the outcome measures. This information has been added to the manuscript. 

2) When reporting the degree of freedom, they should report both the numerator and the denominator

Thank you for pointing this issue out. The numerator and denominator of the degrees of freedom (DFn, DFd) have been added to the tables. 

3) The references have to be homogenized (e.g., l.68: A study by Opie, Evans (22)…)

The references have been edited throughout to improve consistency in formatting.

---

## [Decision Letter · Decision Letter 3]

28 Sep 2022

Effects of short-term arm immobilization on motor skill acquisition

PONE-D-21-19679R3

Dear Dr. Borich,

We’re pleased to inform you that your manuscript has been judged scientifically suitable for publication and will be formally accepted for publication once it meets all outstanding technical requirements.

Kind regards,

François Tremblay, PhD

Academic Editor

PLOS ONE

Additional Editor Comments (optional):

Reviewers' comments:

Reviewer's Responses to Questions

**Comments to the Author**

1. If the authors have adequately addressed your comments raised in a previous round of review and you feel that this manuscript is now acceptable for publication, you may indicate that here to bypass the “Comments to the Author” section, enter your conflict of interest statement in the “Confidential to Editor” section, and submit your "Accept" recommendation.

Reviewer #1: All comments have been addressed

Reviewer #2: (No Response)

2. Is the manuscript technically sound, and do the data support the conclusions?

Reviewer #1: Yes

Reviewer #2: (No Response)

3. Has the statistical analysis been performed appropriately and rigorously? 

Reviewer #1: Yes

Reviewer #2: (No Response)

4. Have the authors made all data underlying the findings in their manuscript fully available?

Reviewer #1: Yes

Reviewer #2: (No Response)

5. Is the manuscript presented in an intelligible fashion and written in standard English?

Reviewer #1: Yes

Reviewer #2: (No Response)

6. Review Comments to the Author

Reviewer #1: The authors already addressed all the responses to my previous comments. I have no further comments to the authors

Reviewer #2: (No Response)

7. PLOS authors have the option to publish the peer review history of their article (what does this mean?). If published, this will include your full peer review and any attached files.

Reviewer #1: **Yes: **Florent Lebon

Reviewer #2: No

---

## [Editor Report · Acceptance letter]

5 Oct 2022

PONE-D-21-19679R3 

Effects of short-term arm immobilization on motor skill acquisition 

Dear Dr. Borich:

I'm pleased to inform you that your manuscript has been deemed suitable for publication in PLOS ONE. Congratulations! Your manuscript is now with our production department. 

Kind regards, 

on behalf of

Dr. François Tremblay 

Academic Editor

PLOS ONE